# Climate change as a driver of insect invasions: Dispersal patterns of a dragonfly species colonizing a new region

Diego Gil-Tapetado[1]*, Diego López-Collar[1], José F. Gómez[1], José Mañani-Pérez[1], Francisco J. Cabrero-Sañudo[1], Jesús Muñoz[2]*

1 Facultad de Ciencias Biológicas, Departamento de Biodiversidad, Ecología y Evolución, Universidad Complutense de Madrid, Madrid, Spain, 2 Real Jardín Botánico (RJB-CSIC), Madrid, Spain

* diego.gil@ucm.es (DGT); jmunoz@rjb.csic.es (JM)

**Data Availability Statement:** All relevant data are within the manuscript and its Supporting Information files.

## Abstract

The dragonfly *Trithemis kirbyi* Sélys, 1891 recently colonized Western Europe from North Africa. Since its first record in the Iberian Peninsula in 2007, the species has been spreading northward and has become naturally established in the central and eastern Iberian Peninsula, the Balearic Islands and southern France. Despite its worldwide distribution, its rapid colonization of the western Mediterranean area occurred only very recently. The aims of this study were to evaluate (1) whether the species' colonization of the western Mediterranean is related to climate change and rising temperatures, specifically the summer warming peaks that have occurred in the last decade, (2) which climatic variables have most influenced its distribution and dispersal, and (3) its potential future dispersal and colonization capacity towards the eastern Mediterranean. We found that the dispersal and recent establishment of *T. kirbyi* in southwestern Europe strongly depends on increasing temperatures, particularly summer temperature peaks, which has allowed this species to disperse farther and more effectively than during years with average summer temperatures. The most important variable in the suitability models is the minimum temperature of the coldest month, which, in recent decades, has become less of a limiting factor for ectotherms. According to the models, suitable areas for the species are currently found throughout the eastern Mediterranean parts of Europe, and it is likely that it can naturally colonize these areas as it did in the Iberian Peninsula. *Trithemis kirbyi* is a model of how climate change and observed rising temperatures have turned previously inhospitable regions into suitable areas for exotic species, which may successfully colonize them naturally if they can reach these promising lands on their own. However, this study serves as a warning that such species can also colonize these new regions with a little help from unsuspecting means, which are often responsible for the increasingly common presence of invasive, noxious taxa in Europe.

## Introduction

Climate change is one of the main drivers of global biodiversity loss [1] owing to its many effects on wildlife, including altering species phenology, physiology, behavior, and distribution

**Funding:** Dr. Diego Gil-Tapetado was supported by a Margarita Salas CT18/22 UCM contract, financed by Universidad Complutense de Madrid with Next Generation funds from the European Union. The funders had no role in study design, data collection and analysis, decision to publish, or preparation of the manuscript.

**Competing interests:** The authors have declared that no competing interests exist.

[2, 3]. Changes in distributions can cause declines in home ranges, resulting in local extirpation processes, habitat loss, and extinction cascades [2], among other issues, which could lead to the need for species conservation measures. Likewise, species that have newly colonized an area may compete with established taxa and these new interactions may generate conservation problems, such as the displacement of native species generally associated with anthropogenically altered habitats [4–6].

Studies on species distribution changes have focused traditionally on groups of animals that are highly conspicuous, such as birds or mammals [e.g., 7–10]. Although monitoring and analyzing spatial distributions may be more complicated in less showy or more poorly recorded taxonomic groups, such as insects, these groups account for most of the "unseen" or "hidden" biodiversity [11], besides of being the most diverse animal group and the largest contributor to biomass on Earth [12, 13].

Among insects, butterflies and dragonflies are used as bioindicators of ecological and climate changes because they are easily recognizable, studied, and monitored [14–16]. Also, given that most species of these two groups can be differentiated and identified by all types of audiences, citizen science platforms can play an important role in helping to record distributions, something for which academic science is finding increasingly difficult to get funded [14, 17, 18]. Here, we focused on the distribution of a species of Anisoptera: *Trithemis kirbyi* Sélys, 1891 (Odonata: Libellulidae) (S1 Fig). The genus *Trithemis* Brauer, 1868, mainly distributed in Africa and Asia, comprises 50 species [19], of which, only four are present in the Western Palearctic [20]: *Trithemis arteriosa* (Burmeister, 1839), *Trithemis festiva* (Rambur, 1842), *Trithemis annulata* (Palisot de Beauvois, 1805) and *T. kirbyi. Trithemis annulata* colonized the Iberian Peninsula at the end of the 1970s by crossing the Strait of Gibraltar from North Africa [21]. Similarly, *T. kirbyi* naturally colonized the Iberian Peninsula from North Africa in 2007 [22] and is locally abundant in this area today [23]. The species also colonized Sardinia and the islands of the Sicilian Channel [24–26], although in lower numbers than in the Iberian Peninsula and with doubts as to whether populations have become established or are dispersing individuals [27].

Over the last couple of centuries, temperature increases have been observed throughout the world [28], including the western Mediterranean (the Mediterranean west of the Italian Peninsula and Sicily). Between 1909 and 1996, the maximum temperature of the southern Iberian plateau increased by 0.71˚C [29]; the aridity of the area also increased during this period [30]. These environmental changes increase the likelihood of colonization of the region by species adapted to warmer and drier conditions, compared with native species, if those species manage to disperse there.

Using species distribution models, we studied the global distribution patterns of *T. kirbyi*, focusing on the western Mediterranean where it has been possible to collect accurate data on the expansion of the species. Our objectives are to analyze (1) whether the species' colonization of the western Mediterranean is related to climate change and rising temperatures, specifically the summer warming peaks that have occurred in the last decade, (2) which climatic variables have most influenced its distribution and dispersal, and (3) its potential future dispersal and colonization capacity in Europe. We hypothesize that *T. kirbyi*, a strong disperser native to areas with warm and dry climates, will accurately track areas that become warmer, colonizing them to establish long-lasting populations that can act as a source of further expansions as the temperature and dryness of the European Mediterranean region increase due to climate change.

## Materials and methods

### Occurrence data

We obtained a global occurrence data set for *T. kirbyi* from GBIF (Global Biodiversity Information Facility, www.gbif.org) and Biodiversidad Virtual (www.biodiversidadvirtual.org), a

citizen science platform with over 2 million georeferenced records for the Iberian Peninsula. Additionally, we cross-checked these data with the Odonata Database of Africa [31, 32] to verify the information for that particular region. Occurrences were then filtered to remove records prior to 1970 and unverifiable data points in water far from the shore. To avoid autocorrelation, only one presence per pixel of the climatic variables at 1 × 1 km was retained. Although we assumed a possible bias towards better sampled areas, such as Europe, than to less sampled areas, such as Central Africa, we kept all the raster cells with occurrences to accurately follow the hypothesized colonization. The final data set included 1,717 occurrences (S2 Fig).

Models were evaluated with occurrences from Odonata Database of Africa, which were not used at all in the modelling procedure.

## Species distribution modelling

We estimated the climate-based potential distribution of *T. kirbyi* using the *biomod2* package [33] and six models: Generalized Linear Model (GLM), Generalized Additive Model (GAM), Artificial Neural Network (ANN), Classification Tree Analysis (CTA), Random Forest (RF), and Maximum Entropy (MaxEnt). The final ensemble model was based on the average of 60 individual models (i.e., 10 iterations × 6 models).

For the bioclimatic variables, we used the 19 variables of the WorldClim 2.1 data set (http://www.worldclim.org) at 1 × 1 km cell size. This data set provides information on seasonality trends, averages, and extreme values of temperature and precipitation from 1970 to 2000, and these bioclimatic variables are widely used to predict the potential distribution of species by reflecting temperature and precipitation patterns and their variation, factors that are overall ecologically meaningful for species [34–36]. We visually inspected the variables and eliminated those with anomalous patterns in the study area, which were mostly due to the difficulty of getting reliable interpolated precipitation values across areas with low precipitation, such as the Sahara Desert or the western Mediterranean [bio 8, bio 9, bio 15, bio 18 and bio 19, see 37]. To establish a set of uncorrelated climatic variables, we intersected the remaining variables with 10,000 randomly selected points in the extent of the study area, performed a correlation analysis, and removed one of the variables from each pair with a Pearson correlation value > 0.7 (S3 Fig), leaving the one perceived to have higher relative ecological importance and predictor effectiveness. The final data set included Mean Diurnal Range (bio2), Isothermality (bio3), Maximum Temperature of Warmest Month (bio5), Minimum Temperature of Coldest Month (bio6), Precipitation of Driest Month (bio14), and Precipitation of Wettest Quarter (bio16).

The construction of background and pseudoabsences was based on a previous simple environmental coverage model with only occurrences, performed with the range between the maximum and minimum values of each bioclimatic variable [38]. Areas that had all their values within the maximum and minimum range of each variable were considered habitable for *T. kirbyi* and were used to establish the background points, whereas areas that did not fulfil at least two of these variables were used to establish pseudoabsence points [39, 40]. During the development of this preliminary model, we observed which bioclimatic variables discriminated the greatest amount of area between habitable and non-habitable areas (i.e., areas outside the maximum and minimum range) and considered them in the final selection of variables (as those that barely discriminate between habitable and non-habitable areas provide less information in the subsequent distribution model).

Presence, pseudoabsence, and background data were split into 75% training and 25% testing data sets to generate an extrinsic Area Under the receiver operating characteristic Curve (AUC) evaluation for the final models that was independent of the intrinsic AUC evaluation of

each individual model generated by *biomod2*. A final ensemble model was obtained by a weighted averaging of the six models in each replica. Weights for each AUC model were calculated, using only those with an AUC > 0.7 [41]. Finally, the ensemble models were evaluated through the extrinsic AUC test with the 25% testing data sets. The continuous ensemble model was binarized to a presence/absence model using the cut-off threshold (0.619) that maximized the True Skill Statistic (TSS). A *post hoc* evaluation of the final model on Africa was carried out using the occurrences of *T. kirbyi* included in the Odonata Database of Africa [31], which were not used in the training or testing data sets used for model generation. For this purpose, we inspected how many points (and their suitability) were either inside or outside the presence/absence threshold suitability value. We selected a number of pseudoabsences equal to the number of occurrences in Africa (828) to perform an AUC test and repeated the process 100 times to obtain an average AUC value to evaluate the ensemble model for Africa.

Variable selection and species distribution models were performed in the R 3.5.0 and RStudio 1.1.453 environments. ArcGIS for Desktop 10.3 (www.esri.com) was used to generate background and pseudoabsence points and model maps.

### Distribution and dispersal analyses

To assess the environmental characteristics of the distribution area of *T. kirbyi* in the western Mediterranean, we converted the continuous suitability values to a presence/absence raster using the cut-off threshold value that maximized TSS (0.619). At each level, we generated 10,000 random points to run ANOVA tests and generate boxplots using presence/absence as a dependent variable. We did not perform this analysis nor the subsequent ones at a global scale due to the high environmental variability of the global study area (e.g., the desert areas of the Sahara can bias the models).

We also studied which variables were most important for the suitability of *T. kirbyi* in the colonized area. The relative importance of each variable and algorithm was calculated by the *biomod2* package. We also performed an independent conditional Random Forest analysis (2,500 trees, 4 variables) with the 20,000 presences and absences previously generated, followed by a simple regression analysis between the resulting most important variables and suitability to assess the significance effect ($R^2$).

To assess the role of summer temperature peaks on dispersal speed and spread, we used the annual mean temperature change with respect to the data from 1880 to 1920, i.e., the annual thermal anomaly, using data at the latitude of the Iberian Peninsula [42]. We calculated annual and cumulative increases in the distribution area of the species (as minimum convex polygons) and compared them against the annual and cumulative thermal anomalies for each year since 2010 (the scarcity of observations between 2007 and 2009 precluded the inclusion of these years) using the Mann–Whitney–Wilcoxon U test. We use both the area (raw variable) and the transformed (linearized) area to perform these analyses.

Finally, we calculated the maximum observed dispersal distance, defined as the absolute distance between the highest latitude point(s) of consecutive years, since the northernmost point(s) can be interpreted as the colonization avant-garde of this dragonfly. We chose from 1 to 5 points for each pair of consecutive years to calculate the dispersal distance mean, standard deviation and, finally, the total dispersal distance for 2010–2022.

### Results

The ensemble model for *T. kirbyi* shows that it has a clear affinity for Mediterranean climates, such as those found in southern Africa or around the Mediterranean Sea basin. According to our results, high suitability areas for this dragonfly are found in Africa, Asia (Arabian

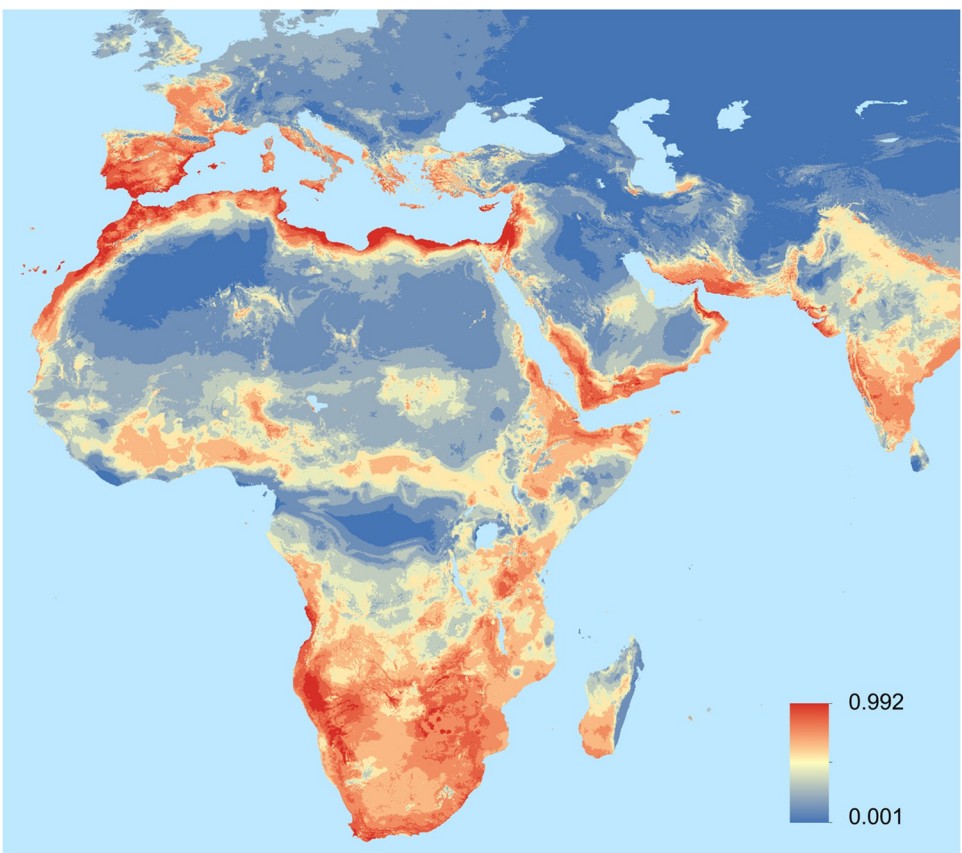

**Fig 1. Worldwide habitat suitability model of *Trithemis kirbyi*.** The ensemble model is based on the average of 60 individual models of six algorithms. Areas showing high suitability values are shown in red, and those of low suitability values, in blue. Software: ArcGIS 10.8.

Peninsula and India), and Europe (Fig 1). In Africa, these areas are limited to mainly southern Africa, specifically in the coastal areas of South Africa and Angola, as well as in Namibia, Eswatini, Zambia, Zimbabwe, some regions of Botswana, the African Great Lakes, the Rift Valley (inner areas of Tanzania and Kenya), and the Horn of Africa. Other high suitability areas of Africa include the mountain areas of central Sahel, the coasts of Morocco, Algeria, Tunisia, Libya, and Egypt, and the eastern islands of the Canary Archipelago. In Western Asia, suitability is high in southeastern Saudi Arabia, Oman, Yemen, the coast of Iran and other Near Eastern countries, and India. In Europe, high suitability areas include the southern half of Portugal, southern, eastern and northeastern Spain, southern and western France, and scattered areas in Italy, Greece, and Turkey, as well as all the Mediterranean islands.

The evaluation of the ensemble model resulted in an AUC of 0.957, indicating an excellent discrimination capacity of the model (S4 Fig). *Post hoc* validation with fully independent data from the Odonata Database of Africa (which were not included in the models used to estimate the global habitat suitability of *T. kirbyi*) also showed satisfactory model predictions (AUC = 0.891).

According to boxplots between selected bioclimatic variables of the ensemble model and presence and absence areas in the Iberian Peninsula, *T. kirbyi* is distributed in areas with high temperatures, low precipitation but with a minimum of rainfall, since the species requires water bodies to complete its life cycle, and high precipitation seasonality (Fig 2). Both the

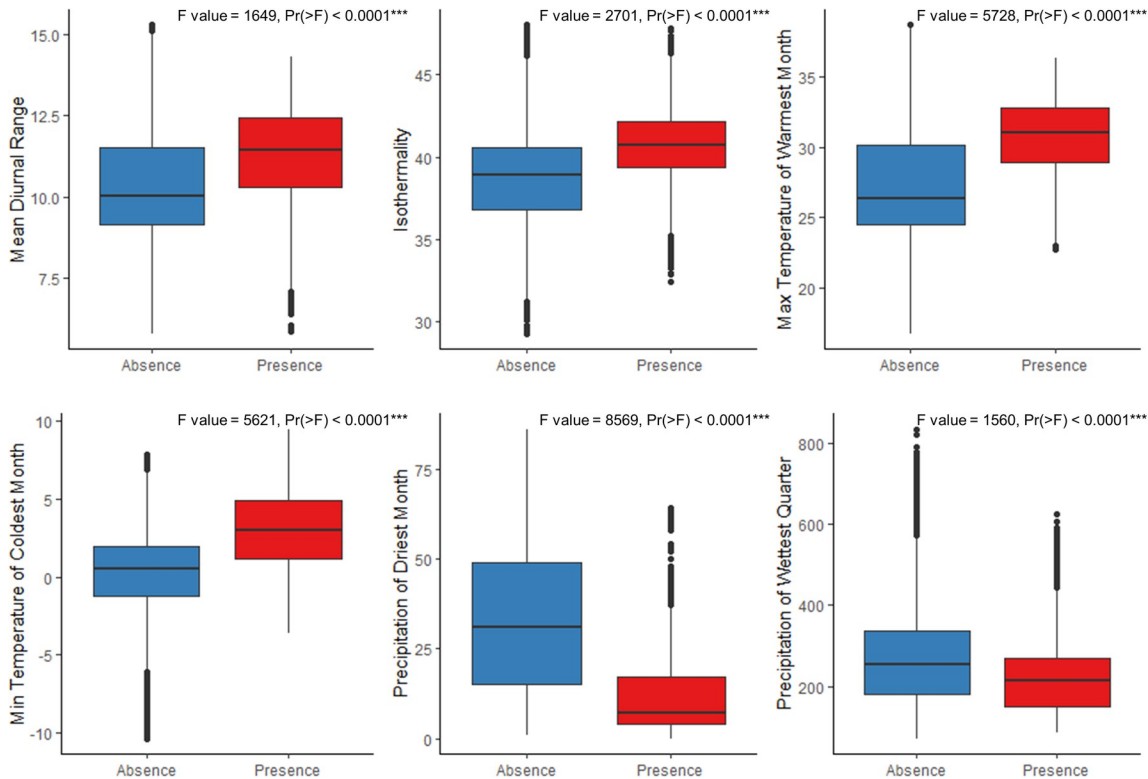

**Fig 2. Boxplots of absence and presence points and variables used in the ensemble model of *Trithemis kirby*, focusing specifically on the Iberian Peninsula and Balearic Islands.** F and Pr(>F) values are included. *** = p < 0.0001. Software: RStudio 1.1.453, under R programming language version 3.5.0.

*biomod2* and the conditional Random Forest analyses indicate that the most important variable in the model is the minimum temperature of the coldest month (bio6, Table 1 and S5 Fig). This variable is strongly correlated with the annual mean temperature (bio1, see S3 Fig) from which the annual and cumulative thermal anomalies were calculated. The linear regression of these two variables with the suitability in the Iberian Peninsula shows a significant positive relationship (suitability ~ bio6: $R^2$ = 0.423, p < 0.00001; suitability ~ bio1: $R^2$ = 0.624, p < 0.00001) (Fig 3).

**Table 1. Contribution of the selected bioclimatic variables to each modelling algorithm as a percentage, and the mean and standard distribution (SD).** Acronyms of the algorithms: GLM, generalized linear model; ANN, artificial neural network; CTA, classification tree analysis; RF, Random Forest; MAXENT, maximum entropy; GAM, generalized additive model. Selected WorldClim bioclimatic variables: bio2, mean diurnal range; bio3, isothermality; bio5, maximum temperature of warmest month; bio6, minimum temperature of coldest month; bio14, precipitation of driest month; and bio16, precipitation of wettest quarter.

| Algorithm | | Variables | | | | | |
|---|---|---|---|---|---|---|---|
| | | bio2 | bio3 | bio5 | bio6 | bio14 | bio16 |
| | GLM | 47.35 | 86.05 | 89.60 | 80.95 | 21.30 | 3.90 |
| | ANN | 55.80 | 36.95 | 26.05 | 77.45 | 24.25 | 30.60 |
| | CTA | 4.55 | 27.00 | 27.15 | 28.70 | 10.35 | 42.45 |
| | RF | 17.10 | 45.50 | 22.80 | 34.65 | 18.25 | 32.10 |
| | MAXENT | 2.45 | 52.40 | 31.65 | 19.35 | 9.85 | 26.10 |
| | GAM | 48.45 | 69.30 | 59.85 | 72.60 | 39.80 | 13.70 |
| | Mean | 29.28 | 52.87 | 42.85 | 52.28 | 20.63 | 24.81 |
| | SD | 23.99 | 21.67 | 26.57 | 27.64 | 11.03 | 13.86 |

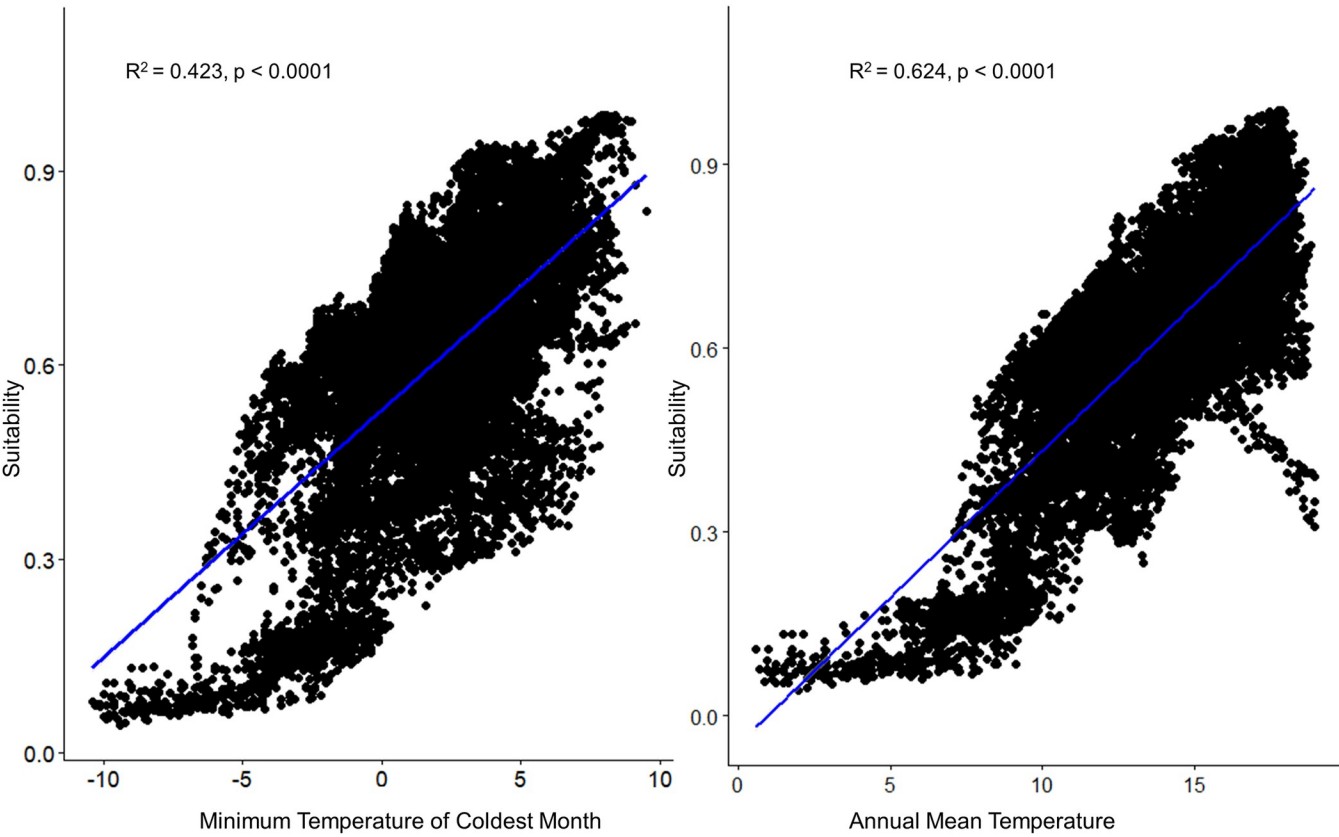

**Fig 3. Linear regressions of the suitability of the ensemble model of *Trithemis kirbyi* and the most important variable, the minimum temperature of the coldest month (bio6), focusing specifically on the Iberian and Balearic area.** The annual mean temperature was also analyzed as this variable is strongly correlated with bio6 and explicitly indicates the positive relationship between dispersal of *T. kirbyi* and high temperatures. Software: RStudio 1.1.453, under R programming language version 3.5.0.

The reconstruction of annual dispersal waves shows that, in the span of a decade, *T. kirbyi* expanded from southern to northern Iberian Peninsula and crossed the Pyrenees into France (Fig 4 and S6 Fig) (the first records for Spain and France were in 2007 and 2017, respectively). The total distance between the first Iberian record and the current northernmost record in France is approximately 1,265 km, suggesting an average annual dispersal of 126 km/year as the crow flies (i.e.: the most direct path between two points). Our observational data show that the mean annual distance travelled in the S–N axis is 429.85 km/year (SD = 199.40). In the Iberian Peninsula, *T. kirbyi* has successfully colonized the entire Mediterranean coast and part of the southern plateau but has not yet colonized the coastal areas of Portugal or the northern plateau. It also has not yet colonized the northern Eurosiberian area (Fig 1). A linear pattern of increase in annual cumulative dispersal is observed from 2010 to 2021, however, this growth plateaus from 2021 to the beginning of 2022 (Fig 5). Yearly increases are observed in the occupied area with peaks occurring during the periods 2012–2013 and 2016–2017. These increases are strongly and positively correlated with the annual thermal anomaly (Mann–Whitney–Wilcoxon U test: area: Z = 2.8241, p = 0.0047, linearized area: Z = 4.1307, p < 0.0001), and also with the difference in thermal anomaly in consecutive years, which peaked during the following periods: 2012–2013, 2015–2016, and 2020–2021 (Mann–Whitney–Wilcoxon U test: area Z = 2.8451, p = 0.0044, linearized area: Z = 3.2649, p = 0.0011).

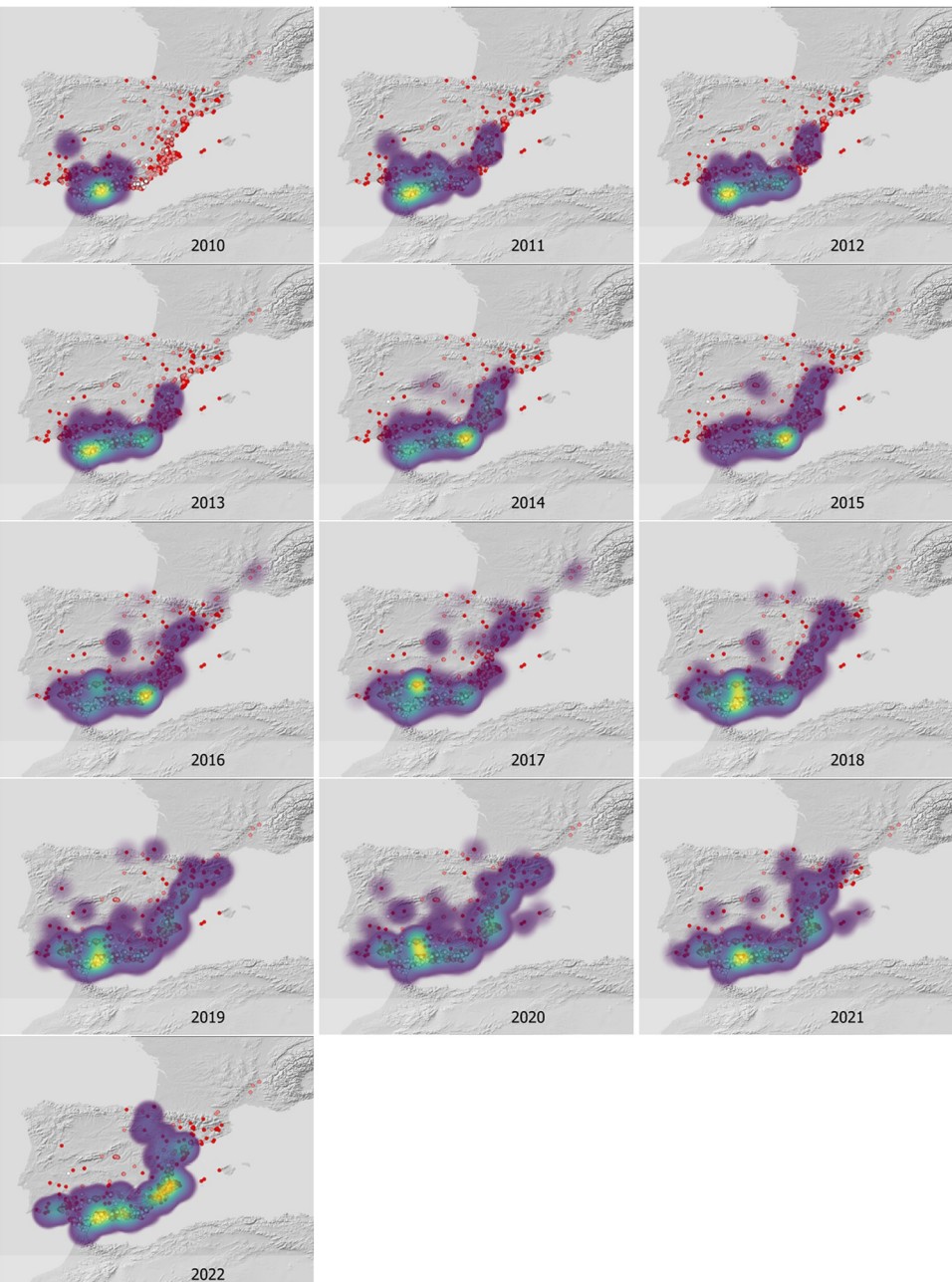

**Fig 4. Evolution of colonization in the Iberian Peninsula between 2010 and spring 2022.** The color of the collection points indicates the relative time the sites were sampled with white dots representing older sites and dark red dots, the most recent ones. The heat map was generated exclusively with the presences of the year represented, and indicates a higher concentration of presences collected in the year in question. Software: ArcGIS 10.8. DEM source: USGS EROS Archive—Digital Elevation—Shuttle Radar Topography Mission (SRTM) Void Filled. Available from: https://earthexplorer.usgs.gov/.

## Discussion

Our study demonstrates that the capability of *T. kirbyi* to colonize new territories is strongly influenced by episodic peaks in maximum temperature (Fig 5), which are becoming more common due to climate change. We also show that summer temperature peaks have a strong

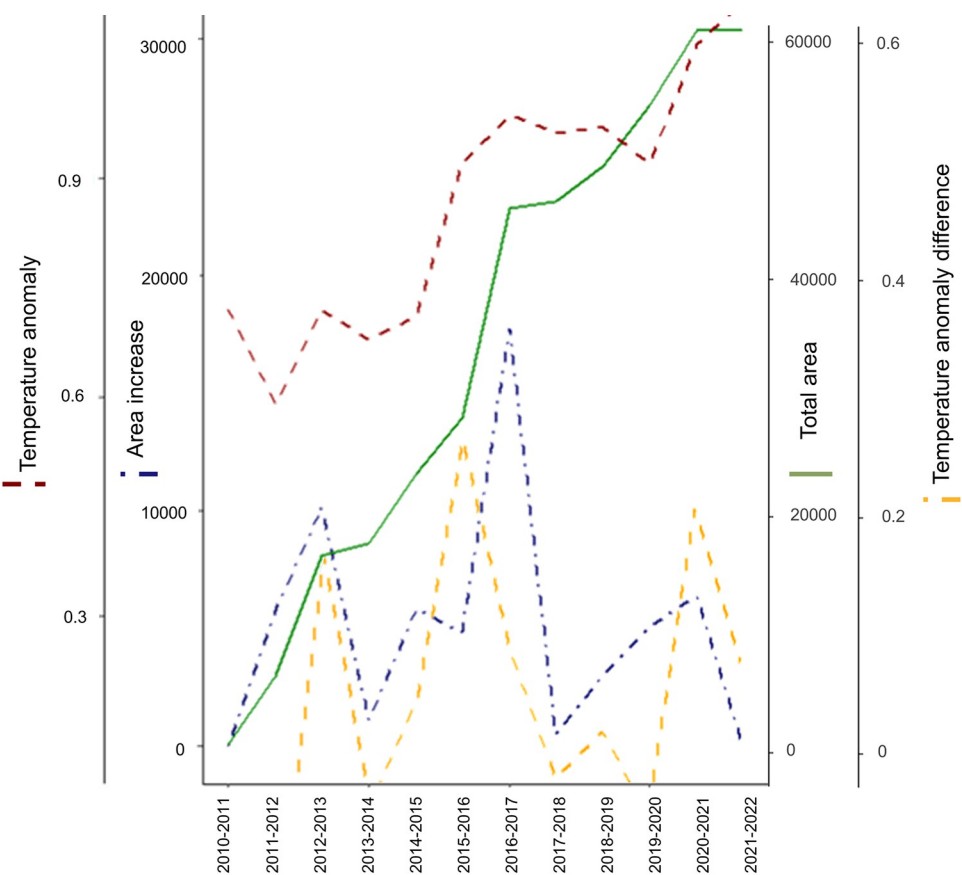

**Fig 5. Trend and comparison of the cumulative distribution area of *Trithemis kirbyi* per year (Total area, green line), the difference/increase in distribution area with respect to the previous year (area increase, blue dashed line), the total annual thermal anomaly with respect to the 1880–1920 period (red dashed line), and the difference/increase of the thermal anomaly with respect to the previous year (yellow dashed line).** Software: RStudio 1.1.453, under R programming language version 3.5.0.

positive effect on dispersal distances travelled, which has allowed a wide expansion towards northern areas that were previously unsuitable for the species. The most important bioclimatic variable explaining the distribution of suitable areas is the minimum temperature of the coldest month (bio6), which, in recent decades, has become less of a limiting factor for ectotherms. Suitable areas for the species can now be found throughout the eastern Mediterranean areas of Europe, making it likely that *T. kirbyi* will naturally colonize them as *T. annulata* did in previous decades.

## Potential distribution and dispersal of *T. kirbyi*

*Trithemis kirbyi* thrives in three biogeographical realms, the Palearctic, Afrotropical and Indomalayan [43], covering a wide distribution in Africa and Eurasia. Its affinity for warm and dry climates makes the western shores of the Mediterranean Sea an ideal location for its northward expansion into hitherto vacant areas that experience an increase in temperature (Fig 3) and dryness. Rapidly rising temperatures, with never-before-seen peaks, may permit this northward expansion to occur at an unprecedented rate (Fig 5).

Although *T. kirbyi* shows a wide but scattered distribution, the model shows areas of maximum suitability connecting the known populations. The Indian area, considered as the

original distribution area of the species, is connected to Africa through the coast of Pakistan, Iran, and the Arabian Peninsula. From this last region, the species' distribution area branches towards southern Africa through the Rift Valley, and towards West (tropical) Africa through an almost continuous high suitability belt. According to the model, the North African coast, the Iberian Peninsula, and the remaining Mediterranean coastal areas are also highly suitable for the species. Although the Sinai Peninsula and coastal areas of the eastern Arabian Peninsula are connected to high suitability areas in the southern Arabian Peninsula, *T. kirbyi* has not been recorded in the region from the Sinai Peninsula to Tunisia except for an isolated locality in Egypt [20]. Thus, an African-European expansion through the Tibesti and Hoggar mountains, as implied by Boudot and Kalkman (20), or through the west coast of Africa, as suggested by our suitability model and the presence of the species in scattered localities in Mauritania, seems more plausible than a colonization from the east (e.g., Anatolian Peninsula), as supported by the lack of records from southeastern Europe. A west African expansion into Europe would also fit with the documented western to eastern colonization of the Mediterranean islands [24, 25].

*Trithemis kirbyi* has a high dispersal potential and can disperse an average of 430 km per year. A mark-recapture study of dragonflies that has been carried out by the Biodiversity Monitoring Group of the Universidad Complutense de Madrid in a peri-urban area of Madrid (Spain) since 2013 supports the long range dispersal capacity of the species. In 2015, several individuals of *T. kirbyi* were marked in this long-term study, representing the first record of this dragonfly in the central Iberian Peninsula [44]. One individual that was first captured and marked in Madrid city (40.4467N 3.7254W) on July 7 was observed approximately 38 km away in La Pedriza (40.7721N 3.8687W) on July 28 (Victor Salvador, personal communication). Its high dispersal capacity is also supported by the fact that *T. kirbyi* colonized the Balearic Islands from the Iberian Peninsula, demonstrating that it can cross great distances over the sea: *T. kirbyi* reached the island of Ibiza from the Iberian Peninsula (127 km) and subsequently Mallorca, where it was observed in a locality 130 km from the nearest occurrence in Ibiza. The species crossed similar distances (100–150 km) during its colonization of the Sicilian Channel islands from the mainland or among the islands [24, 27], as happened in Sardinia in 2003 [26].

## Colonization of an exotic species driven by climate change

*Trithemis kirbyi* is an exotic species in Europe and North Africa [45] that has naturally colonized the Iberian Peninsula through Africa, and is currently expanding into northern European territories. Species occupy areas that fulfil their abiotic (e.g., minimum temperatures that influence the activity of the species) and biotic (presence of water bodies to complete their life cycle) requirements, and that can be reached [the "BAM diagram", 46, 47]. Human-facilitated dispersal combined with climatic niche shifts [48] are responsible for the introduction and expansion of most invasive alien species. In the case of *T. kirbyi*, however, there is not an *a priori* niche shift: the species occupies the same climatic and environmental niche reported for the African populations (such as Mediterranean climate areas as the south of South Africa or North Africa). It also differs from other invasive alien species by being a strong disperser, capable of migrating to areas far from well-established populations. Climate change is generating ideal conditions for *T. kirbyi* in southern Europe, and its dispersal pattern reveals its northern expansion in relatively well-defined waves that coincide with thermal anomaly peaks (Fig 5). This is an example of how previously unsuitable areas have become potentially colonizable or invadable [49], and how climate change can favour biological invasions [50], in addition to natural colonization following successful dispersal events. Further studies could investigate this niche conservatism of the *T. kirbyi* to reinforce our hypotheses.

Case studies of dispersion and establishment in insects are scarce and rely on accurate monitoring of biodiversity [51–54]. Despite the importance of some insect species as agricultural pests or disease vectors, their monitoring is generally not a high priority for governmental administrations. Citizen science platforms such as Biodiversidad Virtual, Observation (https://observation.org), and iNaturalist (https://www.inaturalist.org), however, have proven highly useful for gathering up-to-date, expert-validated information on species introduced into a new territory, and can serve as monitoring systems that can be used by both academic researchers carrying out scientific studies and policymakers responsible for implementing health and environmental measures [55–58].

## Conservation implications

*Trithemis kirbyi* has been able to colonize areas in the Iberian Peninsula that have experienced a high increase in temperature and aridity. This region represents one of the most arid in the southwestern Mediterranean, and models predict it will be severely affected by climate change [59]. Although the species is thermophilic and a generalist, which facilitate its colonization of warmer and drier areas with respect to its abiotic requirements, increased aridity implies a lower availability of water masses, a resource this species needs to complete its life cycle. Changes in water mass availability in increasingly arid environments will impact the future distribution of the species, as is being currently observed with its increasing scarcity in the Sahara Desert.

Rising temperatures caused by climate change have been identified as a factor leading to the expansion of some insect distributions [e.g., 60]. *Trithemis annulata* is another example of a dragonfly expanding into Europe following temperature increases in the 1990s [20], much like our model of *T. kirbyi* [61]. The impacts of these two exotic species of *Trithemis* on native communities have not been studied so far, thus we do not know if they could displace native species or if they are simply new additions to existing plastic and generalist communities. So far, there is no evidence to suggest that *T. annulata* and *T. kirbyi* have become invasive species in the western Mediterranean, but they certainly have become naturalized. Again, citizen science platforms can play a crucial role in detecting the potential impact of these species on native fauna almost in real time [62]. In fact, during the writing of this manuscript, *T. kirbyi* was also found in Belgium during a high temperature episode, reinforcing the idea of the importance of this type of platforms and data, as well as evaluating and validating our model (suitability of this area of Belgium = 0.6) [63].

## Final remarks

The steady and marked increase in temperature since the 1980s, combined with extremely intense heat waves in the last decade [28, 64], has turned a hitherto inhospitable 'cold' region into a new colonizable area with ideal conditions for alien species from southern latitudes. In the last decade, several North African arthropods have been detected in southern areas of the Iberian Peninsula, including *Xylocopa pubescens* Spinola, 1838 [65], *Azanus ubaldus* (Stoll, 1782) [66], and *Cheilomenes propinqua* (Mulsant, 1850) [67]. Although these species colonized the area naturally, similar to *T. kirbyi*, many exotic species have been introduced into Europe from areas much farther away owing to the international trade in goods and inefficient quarantines [68]. We demonstrate natural colonization by an insect that has, so far, shown no signs of harm to natural autochthonous ecosystems; however, our study shows that the road is paved for large-scale invasions that will cost money for the direct control measures they should trigger (e.g., measures against agricultural pests and vector-borne diseases). They will also cause large economic losses due to a decline in biodiversity complexity and associated ecosystem services (e.g., pollination or quality in freshwater ecosystems).

## Supporting information

**S1 Fig. Photograph of a male of *Trithemis kirbyi* Sélys, 1891, in Madrid (Spain).** Photo credit: Jose Ignacio Pascual, with the author's permission. Reprinted from Biodiversidad Virtual (https://www.biodiversidadvirtual.org/insectarium/Trithemis-kirbyi-Selys-1891-img769288.html) under a CC BY license, with permission from Biodiversidad Virtual, original copyright José Ignacio Pascual, 2015.
(TIF)

**S2 Fig. Map showing the location of the occurrence points of *Trithemis kirbyi* included in the analyses.** Software: ArcGIS 10.8.
(TIF)

**S3 Fig. Dendrogram obtained from the cluster analysis used to select the relevant World-Clim variables (in green boxes); the red horizontal line indicates the chosen distance-threshold to form the clusters (i.e., 0.3 or less than 70% correlation). Software: RStudio 1.1.453, under R programming language version 3.5.0.**
(TIF)

**S4 Fig.** Left: Accumulation of the number of presences (red continuous line) and pseudoabsences (blue dashed line) (in y-axis) considering the suitability of the ensemble model of *Trithemis kirbyi* without the population density variable (in x-axis). Center: Boxplots of presences (in red) and pseudoabsences (in blue) considering the suitability of the ensemble model of *Trithemis kirbyi* without the population density variable (in y-axis). Right: Representation of the AUC and AUC value of the *Trithemis kirbyi* ensemble model without the population density variable. Software: RStudio 1.1.453, under R programming language version 3.5.0.
(TIF)

**S5 Fig. Conditional Random Forest analysis (2500 trees, 4 variables) with presences and absences obtained from an *a posteriori* test performed to identify the variables most influential on the distribution of *Trithemis kirbyi*.** Software: RStudio 1.1.453, under R programming language version 3.5.0.
(TIF)

**S6 Fig. Animated version of Fig 4.** Evolution of colonization in the Iberian Peninsula between 2010 and spring 2022. The color of the collection point indicates the relative time since sites were sampled with white dots representing the oldest sites and dark red dots, the most recent ones. The heat map was generated exclusively with the presences of the year represented, and indicates a higher concentration of presences collected in the year in question. Software: Arc-GIS 10.8. DEM source: USGS EROS Archive—Digital Elevation—Shuttle Radar Topography Mission (SRTM) Void Filled. Available from: https://earthexplorer.usgs.gov/. Animation generated at https://www.easygifanimator.net/.
(GIF)

**S1 Table. Complete set of occurrences of *Trithemis kirbyi*, with sources.**
(XLSX)

**S2 Table. Occurrences of *Trithemis kirbyi* used for modelling, after cleaning S1 Table dataset.**
(XLSX)

**S3 Table. Data of the area-thermal anomaly employed to generate Fig 5.**
(XLSX)

**S1 Appendix. R code to run the analyses.**
(PDF)

## Acknowledgments

To Pablo Refoyo for his modelling and analysis advice, to Jean-Pierre Boudot for the shared records of the Odonata Database of Africa records and to Silvio Gómes Fritz for his great and unconditional support.

## Author Contributions

**Conceptualization:** Diego Gil-Tapetado, José Mañani-Pérez, Francisco J. Cabrero-Sañudo, Jesús Muñoz.

**Data curation:** Diego Gil-Tapetado.

**Formal analysis:** Diego Gil-Tapetado, Jesús Muñoz.

**Funding acquisition:** Diego Gil-Tapetado, Jesús Muñoz.

**Investigation:** Diego Gil-Tapetado, José Mañani-Pérez, Jesús Muñoz.

**Methodology:** Diego Gil-Tapetado, Jesús Muñoz.

**Project administration:** Diego Gil-Tapetado.

**Resources:** Diego Gil-Tapetado, José Mañani-Pérez, Francisco J. Cabrero-Sañudo, Jesús Muñoz.

**Software:** Diego Gil-Tapetado, Jesús Muñoz.

**Supervision:** José F. Gómez, Francisco J. Cabrero-Sañudo, Jesús Muñoz.

**Validation:** Diego Gil-Tapetado, Jesús Muñoz.

**Visualization:** Diego Gil-Tapetado, Jesús Muñoz.

**Writing – original draft:** Diego Gil-Tapetado, Diego López-Collar, José F. Gómez, José Mañani-Pérez, Francisco J. Cabrero-Sañudo, Jesús Muñoz.

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
