## [Decision Letter · Decision Letter 0]

3 Aug 2023

PONE-D-23-22273Climate change as a driver of insect invasions: Dispersal patterns of a dragonfly species colonizing a new regionPLOS ONE

Dear Dr. Muñoz,

Thank you for submitting your manuscript to PLOS ONE. After careful consideration, we feel that it has merit but does not fully meet PLOS ONE’s publication criteria as it currently stands. Therefore, we invite you to submit a revised version of the manuscript that addresses the points raised during the review process.

We look forward to receiving your revised manuscript.

Kind regards,

Tzen-Yuh Chiang

Academic Editor

PLOS ONE

Journal Requirements:

4. Please note that funding information should not appear in any section or other areas of your manuscript. We will only publish funding information present in the Funding Statement section of the online submission form. Please remove any funding-related text from the manuscript.

5. We note that Figure 1 in your submission contain map/satellite images which may be copyrighted. All PLOS content is published under the Creative Commons Attribution License (CC BY 4.0), which means that the manuscript, images, and Supporting Information files will be freely available online, and any third party is permitted to access, download, copy, distribute, and use these materials in any way, even commercially, with proper attribution. For these reasons, we cannot publish previously copyrighted maps or satellite images created using proprietary data, such as Google software (Google Maps, Street View, and Earth). For more information, see our copyright guidelines: http://journals.plos.org/plosone/s/licenses-and-copyright.

7. We note that Supporting Information Figures 2 and 6 in your submission contain map/satellite images which may be copyrighted. All PLOS content is published under the Creative Commons Attribution License (CC BY 4.0), which means that the manuscript, images, and Supporting Information files will be freely available online, and any third party is permitted to access, download, copy, distribute, and use these materials in any way, even commercially, with proper attribution. For these reasons, we cannot publish previously copyrighted maps or satellite images created using proprietary data, such as Google software (Google Maps, Street View, and Earth). For more information, see our copyright guidelines: http://journals.plos.org/plosone/s/licenses-and-copyright.

a. You may seek permission from the original copyright holder of Supporting Information Figures 2 and 6 to publish the content specifically under the CC BY 4.0 license.  

Reviewers' comments:

Reviewer's Responses to Questions

**Comments to the Author**

1. Is the manuscript technically sound, and do the data support the conclusions?

Reviewer #1: Yes

Reviewer #2: Yes

2. Has the statistical analysis been performed appropriately and rigorously? 

Reviewer #1: Yes

Reviewer #2: Yes

3. Have the authors made all data underlying the findings in their manuscript fully available?

Reviewer #1: Yes

Reviewer #2: Yes

4. Is the manuscript presented in an intelligible fashion and written in standard English?

Reviewer #1: Yes

Reviewer #2: Yes

5. Review Comments to the Author

Reviewer #1: This is an impressive paper, very professional, outstanding and interesting for a broad audience, and I have no hesitation to recommend publication in its current version and congratulate the authors.

JS CARRION

Reviewer #2: In the manuscript “Climate change as a driver of insect invasions: Dispersal patterns of a dragonfly species colonizing a new region”, Gil-Tapetado and collaborators used citizen-science data to model the potential distribution in Europe (as well as in the native range) of a recently arrived (neonative) dragonfly. They also assessed the potential role of different bioclimatic variables on its survival and spread in Europe and tested for a relation between range expansion and heat waves in the Iberian Peninsula in the period 2010-2022.

They found that this dragonfly will very probably spread along the European Mediterranean coast, as well as inland in Portugal, Spain, France, Italy. Interestingly, they found a relation between the annual rate of increase in the area occupied and the annual temperature anomaly (supposedly the summer heat waves, but see below).

I think that the manuscript is interesting and relevant. Furthermore, it deals with the effects of heat waves on animal spread, a subject that has been difficult to study on terrestrial ecosystems.

I have some major questions, mostly on some methodological issues:

1): Line 196 – specific analysis for the summer temperatures

Here the authors try to assess the role of the summer temperature peaks on dispersal speed and spread. However, they use the mean annual temperature change (annual thermal anomaly). This annual thermal anomaly necessarily includes the effects of mild winters, I guess. Is there in effect a very high correlation between summer temperature anomalies and annual temperature anomalies?

I mean – the authors themselves identified the minimum temperature of the coldest month as the most important variable in their distribution models. The dragonfly dispersal stage is the flying adult, and presumably it will fly in the summer, but I think there should be a discussion on the relative effects of adult flight/ nymph overwinter survival on the pattern of spread.

2) Line 202 –the Mann-Whitney-Wilcoxon U test (a rank sum test) was used to compare increase in area (quadratic) with temperature anomaly (linear). They could have linearized the area. On line 252 these results are referred to as correlations.

3) Line 203 – the explanation of how the authors calculated the maximum observed dispersal distance is unclear. The highest latitude point in consecutive years can occur at widely different longitudes, meaning that a short increase in latitude may refer to a very large or very small distance. So “distance between highest latitude points in consecutive year” is not correct; is it the length of the projection over the same longitude of the northernmost latitudes in consecutive years?

4) Line 315 – “there is no niche shift” – where is this shown? There is a world of models analysing and measuring the niche shifts of invasive species in their new ranges. I don’t think this paper is analysing this issue.

Some more questions, referring to the text:

Introduction – from line 70 onwards, I question the need to build the introduction as if this manuscript would focus on competitive displacement of native species by T. kirbyi. It is out of scope with the goal of the manuscript. I would include the current trends in climate change at the beginning of the introduction, namely mentioning heat waves and current aridification / increase of minimum temperatures / reduction of the harsh conditions in winter

Line 97 – This is of the paragraphs I would move to the beginning of the introduction (it could be the second)

Line 103 - If the point is to present the objectives in this sequence, then rising temperatures and warming peaks must be addressed sooner in the introduction. And a bit more detailed (see the 2 previous comments)

Line 116- what is the study area? The entire distribution of the species whether in its native or neonative range, right? It is not clear throughout the manuscript and should be mentioned in this paragraph

Line 146 – correctly, the authors refer that one of the variables of each pair of correlated variables was removed. But what was the criteria of choice?

Line 172 – a bit confusing and not explained before – so the occurrences of T. kirbyi from the Odonata Database of Africa were not used before? I guess only those occurrences from Africa but retrieved from iNaturalist were used, then? Only on line 225 the authors finally refer that the points from Odonata Database of Africa were not used at all in the modelling procedure.

So, correctly, the authors thought about using this different set of data to validate their models. This was not explained (the use of two separate sets of data for different goals) where it should have been (line 116 - Ocurrence data).

Line 183 – western Mediterranean – the limits of this region are?

Line 238 – Figure 3 is superfluous as a main figure – could be shown as supplemental data

Line 262 – “that were previously unsuitable for the species” – a bit too overinterpreted. In spite of its velocity of expansion, the species could have juts been unable to fly directly to there.

Line 274 – Figure 3? It does not show anything about dryness

Line 343 – “see Figure 2”? This Figure does not show increases in scarcity in the Sahara Desert

Minor issues

Line 60 – “friends” (not necessary)

Line 88 – of which only four (delete “,”)

Line 124 – “than to less”

Line 159 – considered them in

Line 244 - ?as the crow flies?

Line 259 – figure 5?

6. PLOS authors have the option to publish the peer review history of their article (what does this mean?). If published, this will include your full peer review and any attached files.

Reviewer #1: **Yes: **José S Carrión

Reviewer #2: No

---

## [Author Response · Author response to Decision Letter 0]

12 Aug 2023

We have uploaded a "Response to Reviewers.docx" document answering to the points raised in the Decision Letter

---

## [Decision Letter · Decision Letter 1]

25 Aug 2023

Climate change as a driver of insect invasions: Dispersal patterns of a dragonfly species colonizing a new region

PONE-D-23-22273R1

Dear Dr. Muñoz,

We’re pleased to inform you that your manuscript has been judged scientifically suitable for publication and will be formally accepted for publication once it meets all outstanding technical requirements.

Kind regards,

Tzen-Yuh Chiang

Academic Editor

PLOS ONE

Additional Editor Comments (optional):

Reviewers' comments:

Reviewer's Responses to Questions

**Comments to the Author**

1. If the authors have adequately addressed your comments raised in a previous round of review and you feel that this manuscript is now acceptable for publication, you may indicate that here to bypass the “Comments to the Author” section, enter your conflict of interest statement in the “Confidential to Editor” section, and submit your "Accept" recommendation.

Reviewer #2: All comments have been addressed

2. Is the manuscript technically sound, and do the data support the conclusions?

Reviewer #2: (No Response)

3. Has the statistical analysis been performed appropriately and rigorously? 

Reviewer #2: (No Response)

4. Have the authors made all data underlying the findings in their manuscript fully available?

Reviewer #2: (No Response)

5. Is the manuscript presented in an intelligible fashion and written in standard English?

Reviewer #2: (No Response)

6. Review Comments to the Author

Reviewer #2: (No Response)

7. PLOS authors have the option to publish the peer review history of their article (what does this mean?). If published, this will include your full peer review and any attached files.

Reviewer #2: No

---

## [Editor Report · Acceptance letter]

7 Sep 2023

PONE-D-23-22273R1 

Climate change as a driver of insect invasions: Dispersal patterns of a dragonfly species colonizing a new region 

Dear Dr. Muñoz:

I'm pleased to inform you that your manuscript has been deemed suitable for publication in PLOS ONE. Congratulations! Your manuscript is now with our production department. 

Kind regards, 

on behalf of

Dr. Tzen-Yuh Chiang 

Academic Editor

PLOS ONE